# The Small Mobile Ozone Lidar (SMOL): instrument description and first results

Fernando Chouza[1], Thierry Leblanc[1], Patrick Wang[1], Steven S. Brown[2], Kristen Zuraski[2,3], Wyndom
Chace[2,3,4], Caroline C. Womack[2,3], Jeff Peischl[2,3], John Hair[5], Taylor Shingler[5], John Sullivan[6]

[1]Jet Propulsion Laboratory, California Institute of Technology, Wrightwood, California, USA
[2]NOAA Chemical Sciences Laboratory, Boulder, Colorado, USA
[3]Cooperative Institute for Research in Environmental Sciences, University of Colorado Boulder, Boulder, Colorado, USA
[4]Department of Chemistry, University of Colorado Boulder, Boulder, Colorado, USA
[5]NASA Langley Research Center, Hampton, Virginia, USA
[6]NASA Goddard Space Flight Center, Greenbelt, Maryland, USA

*Correspondence to*: Fernando Chouza (keil@jpl.nasa.gov)

**Abstract.** Ozone profile measurements at high temporal and vertical resolution are needed to better understand physical and chemical processes driving tropospheric ozone variability and to validate the tropospheric ozone measurements from spaceborne missions such as TEMPO (Tropospheric Emissions: Monitoring Pollution). As part of the Tropospheric Ozone Lidar Network (TOLNet) efforts allocated to provide such measurements, and leveraging on the experience of more than 20 years of ozone lidar measurements at Table Mountain Facility, the JPL lidar group developed the SMOL (Small Mobile Ozone Lidar), an affordable differential absorption lidar (DIAL) system covering all altitudes from 200 m to 10 km. a.g.l. The transmitter is based on a quadrupled Nd:YAG laser which is further converted into a 289/299 nm wavelength pair using Raman shifting cells, and the receiver consists of three ozone DIAL pairs, including one 266/289 and two 289/299 nm. Two units were deployed in the Los Angeles basin area during the Synergistic TEMPO Air Quality Science (STAQS) and Atmospheric Emissions and Reactions Observed from Megacities to Marine Areas (AEROMMA) campaigns in summer 2023. The comparison with airborne in-situ and lidar measurements show very good agreement, with systematic differences below 10 % throughout most of the measurement range. An additional comparison with nearby surface ozone measuring instruments indicates unbiased measurements by the SMOL lidars down to 200 m above ground level. Further comparison with the Goddard Earth Observing System Composition Forecast (GEOS-CF) model suggests that such lidars are a critical tool to perform model validation and can potentially be used for assimilation to air quality forecasts.

**1 Introduction**

The monitoring of tropospheric ozone is crucial to understand atmospheric chemistry and its impact on human health (US EPA, 2006). It is very challenging for any single technique to be able to address current monitoring requirements because the concentration of tropospheric ozone can fluctuate over small temporal and spatial scales as the result of different factors, including the emission rate of precursors, solar radiation intensity, and advection processes. The recently launched TEMPO mission (Chance et al., 2013; Zoogman et al., 2017) provides outstanding temporal and spatial coverage over the continental United States. Nevertheless, its profiling capabilities are limited when compared with other techniques like lidar. Ground-based ozone lidars, while limited in spatial coverage, have the potential to complement the TEMPO mission with long-term high vertical and temporal resolution measurements as required to understand different processes like regional and long-range transport (Chouza et al., 2021), stratosphere-troposphere exchange, and complex low-level dynamical processes in coastal regions. Furthermore, such ground-based measurements provide a reference and validation capability for TEMPO and future spaceborne instruments (e.g. GeoXO).

Over the last decade substantial progress has been made towards commercially viable and robust lidar systems, with the largest progress concentrated in ceilometers, wind, and water vapor lidars. While some of that progress can be attributed to the leveraging on the development of infrared laser sources by the telecommunications industry, substantial progress can also be credited to system automation techniques (Engelman et al., 2016; Shimizu et al., 2016) and a conscious effort to make them more robust and less costly (Spuler et al., 2015). The Small Mobile Ozone Lidar (SMOL) system is intended to contribute to this effort, providing TOLNet (https://tolnet.larc.nasa.gov, last access: 4 June 2024) with a more cost-effective ozone lidar for air quality monitoring, satellite, and model validation as well as the potential for assimilation for air quality forecasts. While developing the SMOL concept, a few design criteria and requirements were followed:

1. Lidars are typically associated with a large upfront cost. In the case of SMOL, the marginal hardware cost was capped to 100k USD to make it competitive with alternative measurement techniques.

2. Another issue often associated with lidars is the high operational cost due to the need of qualified lidar personnel. A fully autonomous system with limited and simple maintenance needs was an additional requirement to reduce the cost per acquired profile. By having a network of identical systems, we also expect to reduce the processing and data archiving burden by simplifying the processing chain.

3. Finally, the performance of the SMOL system had to be comparable to that of the already existing TOLNet lidar systems, covering the low and mid troposphere with a temporal resolution of 30 minutes or better, a random uncertainty of under 10% and an effective vertical resolution ranging from 100 m to 1 km.

This paper provides an overview of the SMOL system design, as well as the results and lessons learned from the first measurements of two nearly identical units, namely SMOL-1 and SMOL-2. Section 2 presents an overview of the system hardware. The data processing algorithm is reviewed in section 3. Results from the first measurements in the field during the STAQS/AEROMMA campaigns and comparison with co-located measurements are reviewed in section 4. Finally, Section 5 discusses the performance of the system and provides insights for future improvements.

## 2 Instrument Description

The development of the SMOL lidar started in 2021 with the aim to fulfil the need for a lidar capable of reliably providing low-cost ozone measurements in the troposphere. The SMOL design leverages on the lessons learned from earlier attempts to established continued ozone monitoring in the troposphere (Bösenberg, 2000; Trickl et al., 2020), as well as over two decades of tropospheric ozone lidar measurements at JPL Table Mountain Facility (TMF) by the Table Mountain Tropospheric Ozone Lidar (TMTOL) (McDermid et al., 2002) for NDACC (Network for the Detection of Atmospheric Composition Change (De Maziere et al., 2018) and TOLNet.

The SMOL system (Fig. 1) is built around a two-door aluminium enclosure on wheels. The two-door setup allows easy access to all lidar subsystems and facilitates any field maintenance required on the unit, while having the unit on wheels allows the relocation of the unit over short distances and the loading and unloading from pickup trucks without the need for additional equipment like forklifts. The overall dimensions of the unit are 1m x 1.5 m x 2 m (width x depth x height), including a protective barrier added at the top of the enclosure to prevent accidental human exposure to the outgoing laser beams. The weight of the unit is approximately 400 kg. The power requirement of the unit is approximately 2 kW, with a split phase 120/240V L14-30 receptacle being the standard power supply configuration to keep the air conditioning and the rest of the lidar subsystems on different circuits. If a split phase supply is not available, the unit can be reconfigured to operate from a single phase 120V supply.

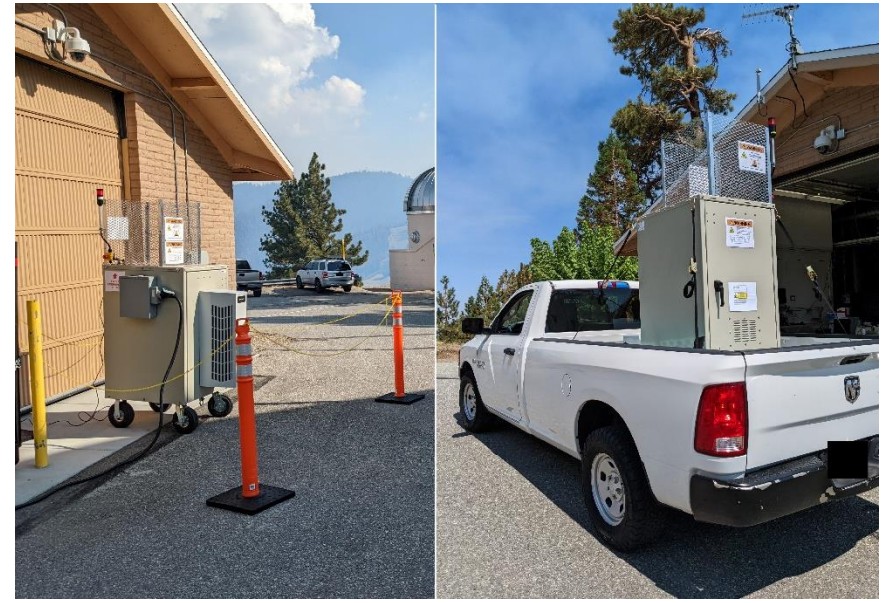

**Figure 1. (Left) SMOL deployed at JPL Table Mountain Facility. (Right) SMOL loaded in a pick-up truck ready for transportation.**

To be able to operate in various environments, a temperature-controlled enclosure is desirable to prevent overheating of the laser, minimize misalignments associated with thermal cycling, and reduce efficiency variations in several components associated with temperature changes. In the case of the SMOL systems, this is accomplished via an air conditioning system

90 with heating and cooling mode attached to one of the enclosure doors. The main specifications of the unit are summarized in Table 1, while further details are provided in the following subsections and Fig. 2.

**Table 1: Main specifications of SMOL. SP stands for short pass.**

| General specifications | |
|---|---|
| Size | 1m x 1.5 m x 2 m (width x depth x height) |
| Weight | 400 kg |
| Power | 2 kW (120/240V L14-30 receptacle) |
| **Transmitter** | |
| Type | Raman conversion in $H_2$ and $D_2$ pumped by a quadrupled Nd:YAG laser |
| Pump laser source | Quadrupled flashlamp-pumped Nd:YAG |
| Repetition rate | 20 Hz |
| Energy per pulse at laser output (266 nm) | 50 mJ |
| Energy per pulse at Raman cell output (266, 299, 289 nm) | 1 mJ, 5 mJ, 5 mJ |
| Raman cell pressure ($D_2$) | 12.4 bar |

| | High intensity | Mid intensity | Low intensity |
|---|---|---|---|
| Raman cell pressure ($H_2$) | 5.5 bar | | |
| Beam divergence at Raman cell output | 0.25 mrad (full angle, $1/e^2$) | | |
| Beam diameter at Raman cell output | 15 mm | | |
| **Receiver** | | | |
| | **High intensity** | **Mid intensity** | **Low intensity** |
| Type | Newtonian | Refractive | Refractive |
| Diameter | 152.4 mm | 25.4 mm | 25.4 mm |
| Focal length | 762 mm | 100 mm | 100 mm |
| Field of view | 1.3 mrad | 4 mrad | 4 mrad |
| Fiber diameter | 1 mm | 0.4 mm | 0.4 mm |
| Optical filters/bandwidth (nm) | 292/32, 285/10, 291.1/1.2 | 292/32, 285/10, 291.1/1.2 | 300 (SP), 266/5, 285/10 |
| Detectors | 2 Hamamatsu H12386-110 | 2 Hamamatsu H12386-110 | 2 Hamamatsu H12386-110 |
| **Signal acquisition** | | | |
| Control computer / Digitizer | Xilinx Zynq-7010 system-on-chip | | |

## 2.1 Transmitter

The SMOL transmitter is based on a flashlamp-pumped Nd:YAG laser followed by doubling and quadrupling crystals. The laser unit outputs 266 nm pulses with an energy of 50 mJ at 20 Hz. The laser output energy is stable to within ± 10% for multiple days without needing readjustment. As the flashlamps deteriorate and power decreases, remote adjustments to the flashlamp voltage and temperature adjustments to the doubling and quadrupling crystals allow to partially offset the power decrease and extend the service intervals. The output of the laser at 266 nm is then divided by a 50/50 beam splitter and redirected by piezo actuated mirrors into two Raman conversion cells filled with hydrogen and deuterium at 5.5 bar and 12.4 bar, to shift the input 266 nm wavelength to 299 and 289 nm, respectively. The piezo actuated mirrors allow to steer the beams and align them to the receivers. The Raman cells have plano-convex lenses with a 250 mm focal length at their input to improve the Raman conversion efficiency. The output of the cells is then recollimated and transmitted through anti-reflection coated fused silica windows into the atmosphere. The output of the system is approximately 5 mJ at 289 nm, 5 mJ at 299 nm, and 1 mJ at 266 nm, which corresponds to a conversion efficiency of approximately 20% at the first Raman Stokes. The output diameter of both beams is 15 mm, with a divergence of 0.25 mrad. The recollimation of the transmitted beam is verified by scanning the beam over the field of view (FOV) of the receivers and looking for an intensity plateau on these scans at given altitude when full overlap and no saturation is expected (>2.5 km for the high range receiver). The angular width of this plateau is a combination of the beam divergence and the receiver FOV. With this configuration, the unit has a NOHD (Nominal Ocular Hazard Distance, 0.25 sec) of ~120 m, which allow operations without restrictions of air-traffic control.

## 2.2 Receiver

The SMOL receiver consists of three fiber-coupled telescopes to accommodate the dynamic range of the atmospheric returns. The high and medium range receivers are setup to receive the backscattered light originating from both Raman cells (289 nm and 299 nm), while the low range receiver is setup to receive the 266 nm and 289 nm wavelengths coming out from the deuterium-filled Raman cell. The 266/289 nm wavelength pair is not only less sensitive to aerosol contamination (Chouza et al., 2019) typically found in the boundary layer, but also corresponds to the output of only one of the cells, thus reducing the sensitivity to transmitter/receiver misalignment in the lowermost part of the receiver range.

The high-altitude receiver is implemented with a 6-inch (152.4 mm) diameter parabolic f/5 mirror coupled into a 1 mm fiber, while the medium and low altitude receivers are built with 1-inch (25.4 mm) diameter lenses (100 mm nominal focal length) focused into 0.4 mm fibers, respectively. All fibers have a numerical aperture of 0.22. The resulting FOV of receivers are 1.3 mrad for the high range and 4 mrad for the medium and low range receivers. While smaller FOVs could help to reduce the impact of solar background on the 299 nm channels during daytime operation (266 nm and 289 nm are practically solar blind), the instrument range is mostly limited by the on-wavelength absorption. Furthermore, such change would make the instrument more sensitive to misalignment caused by temperature changes, vibration, laser pointing jitter, etc.

The fiber outputs of the receiving telescopes are redirected into a spectrometric detection unit, where the output of the fibers is recollimated. After recollimation, the atmospheric backscatter is sent through a first set of filters for additional solar background reduction. In the case of the high and medium range channels, the filter has a center wavelength of 292 nm and a 32 nm full-width at half-maximum (FWHM) transmission window, while for the low range receiver, a short pass filter with a cut-off wavelength of 300 nm is used.

After solar background reduction, the atmospheric backscatter of the high and medium range channels is split into two beams with 50:50 beam splitters. The use of intensity beam splitters instead of dichroic beam splitters means that half of the received signal is discarded, acting effectively as an attenuator. This design decision is based on cost considerations and the fact that additional signal strength would cause detector saturation at lower altitudes, which would require additional receiver sets to accommodate for the lidar range of interest. The 289 nm detection arm uses an interference filter with a 285 nm center wavelength and 10 nm FWHM bandpass window, while the 299 nm detection arm uses a 299.1 nm center wavelength with a 1.2 nm FWHM transmission window. In the case of the low altitude receiver, a dichroic beam splitter is used to separate the 289 and 266 nm returns. The 289 nm receiver arm uses the same filter as in the other two receiver pairs. The 266 nm receiver arm uses an interference filter with the center wavelength of 266 nm and a FWHM transmission window of 5 nm.

Finally, all the interference filters are followed by plano-convex lenses that focus the atmospheric return into the photocathode of photomultiplier tubes (PMTs). All the PMTs used in SMOL are of the photon-counting type (Hamamatsu H12386-110). While these photomultipliers are relatively slow (20 ns pulse pair resolution) compared to other PMTs used by other lidars at JPL TMF (typically 5 ns pulse pair resolution), the fact that they have a built-in discriminator in the same package as the PMT minimizes the chance of electrical noise from the laser and other subsystems to impact the signal. Furthermore, this also

simplifies the design and eliminated the need to further adjust the discriminator level as this is made in the factory to optimize the detector performance. The output of these detectors is sent into a multi-channel scaler (MCS) implemented on a Xilinx Zynq-7010 system-on-chip (SoC), where the signals are digitized and stored (every 3 minutes) in Hierarchical Data Format version 5 (HDF5) together with a set of system parameter needed for the data retrieval (system location, elevation, and bin number corresponding to the zero range).


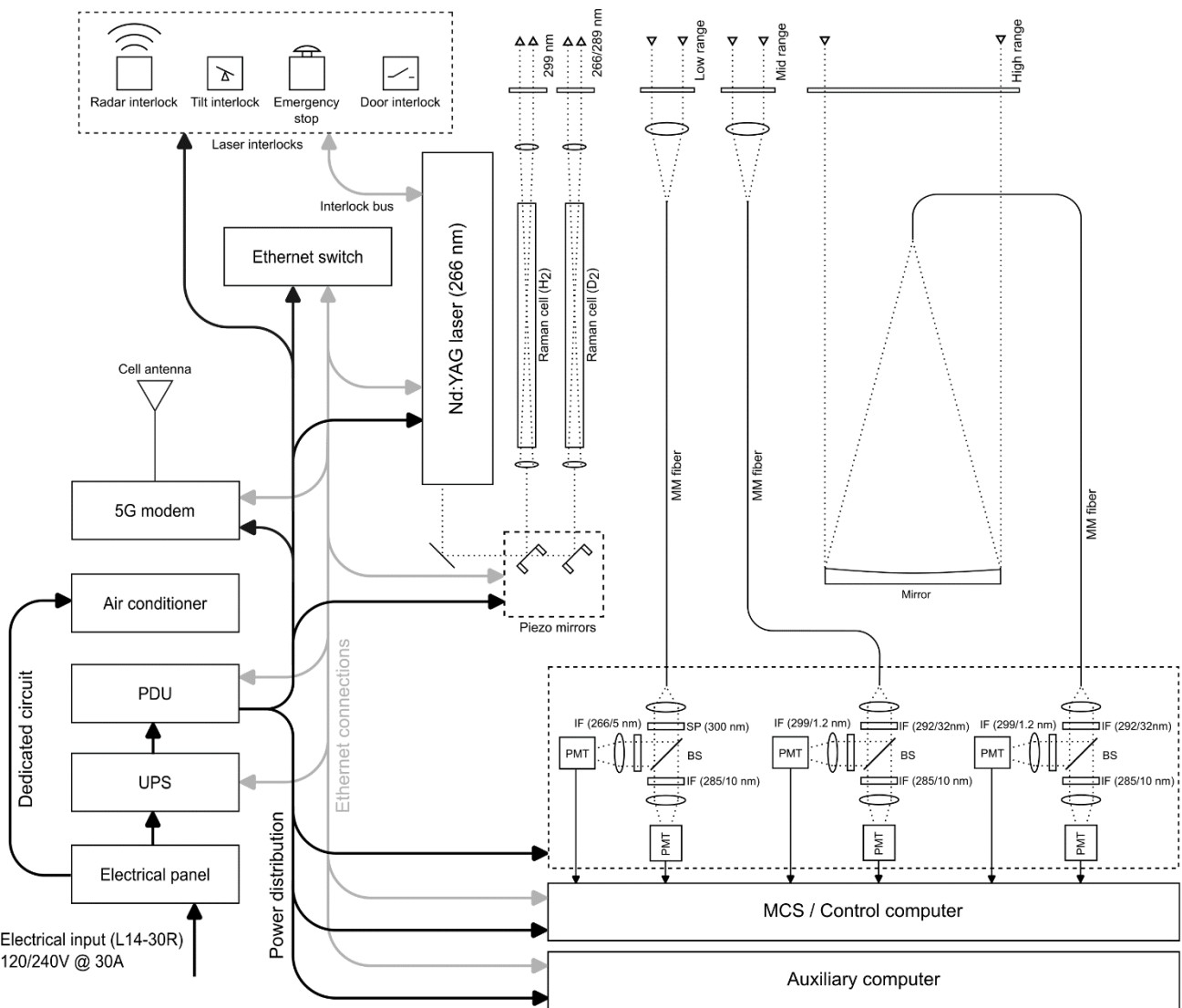

**Figure 2. Schematic of SMOL. MM is multi-mode, SP is short pass, BS is beam splitter, IF is interference filter, MCS stands for multi-channel scaler, PDU is a power distribution unit, and UPS is an uninterruptible power supply.**

## 2.3 System automation and auxiliary systems

The SMOL systems were conceived to operate in a fully autonomous mode based on a pre-loaded schedule that typically ranges from a few hours per day to 24/7 operations. The automation software runs in the same System-on-a-Chip (SoC) as the data acquisition and provides a web-based interface that allows to monitor the system status and data acquisition in real time, similar to what is already implemented in the rest of the JPL-TMF lidar systems (Chouza et al., 2019).

   Whenever the preloaded schedule requires the system to be started, the SMOL controller commands the power distribution
unit to turn on the laser power supply, and via an Ethernet interface, commands the start of the laser. Finally, after a brief warmup period, the automation software starts the signal acquisition. After the prescheduled measurement window is completed, the SMOL automation software stops the acquisition, commands the stop of the lasing, and shuts off the power to the laser power supply.

   Additionally, on a prescheduled basis, the SMOL lidar performs an alignment routine. This alignment routine commands the
piezo-actuated mirrors located at the input of the Raman conversion cells and search for the position that achieves the maximum backscatter signal of the high range receiver at a prescribed altitude (2.5 km a.g.l.). This alignment routine is only intended to compensate for small drifts in the system alignment. If, based on the resulting ozone profile, large misalignments are suspected, a manual realignment has to be conducted.

   In order to protect the system from unexpected power outages, an uninterruptible power supply (UPS) unit is included. This
UPS is able to provide energy to the system in operation (without including the climate control system) for about 10 minutes. If the UPS detects loss of power during a measurement period, the software safely shuts off the laser and sends a notification to the operators.

   The unit air conditioning system is programmed to maintain a temperature of 298K inside the enclosure. If the lidar computer senses that the temperature or humidity inside the enclosure is outside a safe range, it will notify the operators.

Since the unit fully operates in remote mode, the safety engineering controls of the systems are crucial to minimize the hazards associated with the lidar operation. The set of engineering measures to avoid accidental exposure to the laser includes locked access doors equipped with a safety interlock, a mechanical barrier that prevents accidental access to the outgoing laser beams, a tilt sensor that interlocks the laser in case the unit inclination is larger than 15 degrees, an external laser emergency interlock button, and an upward pointing microwave motion sensor that commands a laser shutter system to block the laser beam every
time a moving object approaches the top of the lidar unit. This feature is intended to provide additional safety measurements to prevent boom lift and ladder users working around the lidar to get accidentally exposed to the outgoing laser beams. The sensitivity of the system can be adjusted to prevent most of the false positive detections caused by birds and other smaller targets.

   The communication with SMOL is typically conducted via a 5G cellular modem, with WiFi as an alternative source of
connectivity. A secondary computer directly connected to the modem is included as a safety measure to allow remote restart of different peripherals if the main control computer becomes unresponsive.

## 3 Data Processing

The SMOL raw lidar data (example shown in Fig. 3) acquired in HDF-5 format are processed using the Global Lidar Analysis Software Suite (GLASS) data processor developed in-house at JPL-TMF. The GLASS program is a state-of-the-art lidar processing software written in Interactive Data Language (IDL) and initially developed to retrieve stratospheric ozone, temperature, aerosol, tropospheric ozone, and water vapor for the four JPL lidars contributing to NDACC. GLASS was later expanded to process the raw data of a dozen other lidar instruments contributing to the NDACC, TOLNet and GRUAN (GCOS reference Upper Air Network) networks.

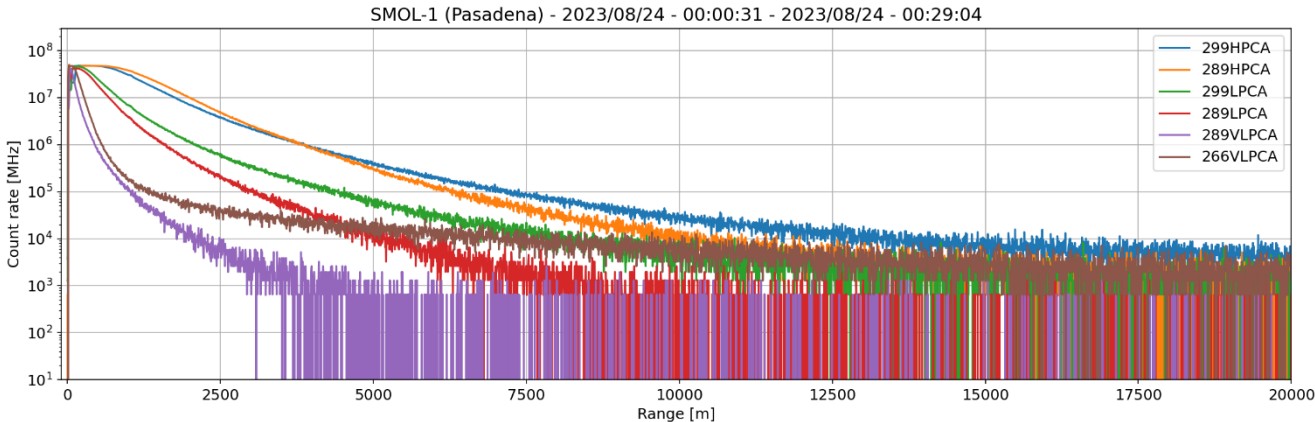

**Figure 3. Raw signals of SMOL-1 during AEROMMA averaged over 30 minutes. The high range receiver signals are 299HPCA (blue) and 289HPCA (orange). The mid-range receiver signals are 299LPCA (green) and 289LPCA (red). The low range receiver signals are 289VLPCA (purple) and 266VLPCA (brown).**

Before ozone is retrieved, the SMOL raw lidar signals are first corrected for non-linearity due to pulse-pileup (saturation), and background noise. To extract background noise, several noise fitting function options are available (constant, linear, polynomial, or a series of one or more exponential functions). For the SMOL systems, background noise most often consists of a combination of constant sky light and PMT dark current but can occasionally include a time-dependent (i.e., altitude-dependent) noise component (signal-induced noise, or SIN). This typically happens when the PMTs are illuminated by light returns from the lower altitudes, becoming more pronounced for shorter wavelengths. The most obvious case of SIN can be seen on the 266 nm channel on Fig. 3. While 266 nm is almost completely solar blind due to the strong ozone absorption at this wavelength, the background on this channel is far from zero and is not constant with range, which is an indication of SIN. In this case, SIN can be modelled as a series of two exponentials.

Over the course of the SMOL development, the magnitude of the signals for each channel (especially the high intensity channels) were refined to optimize the balance between high signal-to-noise ratio (STNR) and low SIN.

After correction, the lidar signals are checked for the presence of particulate layers. For the SMOL instruments, raw signals contaminated by clouds or thick aerosol layers are typically discarded but can be corrected for if the particulate layer does not exceed a specified optical thickness. The correction consists of removing the particulate backscatter interference inside the

cloud layer while ignoring the extinction interference. This method is basic and provides only a single, averaged value of ozone inside a thin cloud or aerosol layer, but it has the advantage of being deterministic and very stable. In their first release (Rapid Delivery version), the SMOL ozone profiles are not corrected for aerosols. In an effort to optimize the ozone product, it is

planned to upgrade GLASS with a state-of-the-art aerosol correction in the near future.

GLASS uses the Differential Absorption Lidar (DIAL) technique first described by Pelon and Mégie (1982), with the derivative step implemented through a Savitsky-Golay (SG) derivative filter followed by a Blackman filter for additional noise reduction. GLASS uses standardized definitions of vertical resolution and uncertainty as described in Leblanc et al. (2016a; 2016b). The uncertainty sources considered include measurement noise (Poisson statistics), absorption cross-sections and their

temperature dependence, molecular extinction, saturation correction, background noise extraction, and aerosol correction (if applicable). The effective vertical resolution scheme used in GLASS can either be constrained by altitude or by random uncertainty. An altitude-constrained vertical resolution scheme consists of fixing vertical resolution as a function of altitude, independently of the lidar STNR. On the other hand, a noise-constrained vertical resolution scheme consists of applying a specific amount of vertical smoothing (controlled by the length of the SG and Blackman filter windows) such that the STNR

after smoothing remains constant. For the SMOL instruments, the default resolution scheme is a constant 7% random uncertainty noise-constrained scheme throughout the profile, but with vertical resolution not exceeding 500-m, 1 km and 1.5 km for the low-intensity, medium-intensity and high-intensity ranges respectively. After ozone is retrieved for each intensity range (low, medium, high), a single merged profile is obtained by selecting the best combination of each range.

The SMOL data processing by GLASS is typically done automatically for a given time interval. The results undergo thorough

QA/QC before they are uploaded to the TOLNet data server (https://tolnet.larc.nasa.gov/download, last access: 4 June 2024). For the SMOL instruments, the default high-temporal resolution product (referred to as "HIRES" product at TOLNet) consists of one profile every 30-minutes. Other products with different temporal or vertical resolutions can be produced, depending on the application needs (e.g., "CLIM", "CALVAL").

## 4 First deployment: SARP, STAQS, and AEROMMA

Several field campaigns focusing on air quality in heavily populated areas of the US took place between the end of June and the beginning of August 2023. As part of the TOLNet contribution to these efforts, the JPL TMF lidar group deployed two SMOL units to the Los Angeles (LA) basin region and operated them in conjunction with the fixed TMTOL system at TMF (34.38° N, 117.67° W). The SMOL-1 unit was deployed at JPL main campus in Pasadena (34.20° N, 118.17° W), while the SMOL-2 unit was deployed at the campus of the California State University in San Bernardino (34.19° N, 117.33° W) (Fig.

4). The selection of the deployment locations resulted from a combination of logistical considerations, campaign aircraft flight planning, co-located ground-based instrumentation availability, as well as wanting to investigate the ozone observed variability at these locations.

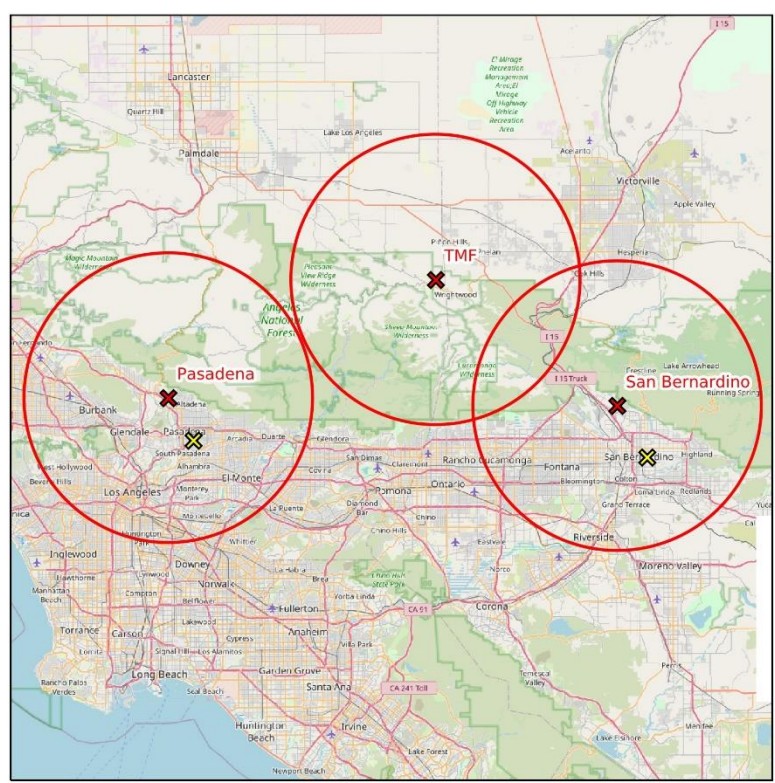

**Figure 4. Map showing the deployment location of the three JPL lidars (red crosses), surface measurements (yellow crosses), and co-location radius (25 km) used for the comparison with airborne in-situ and lidar (red circles). Map Data © OpenStreetMap Contributors**

The measurements during these campaigns were mainly grouped into two intensive observation periods (IOPs) of approximately 5 days each, with some measurement in-between. The first IOP took place between 25 June and 29 June 2023

and was coincident with the deployment of the NASA DC-8 airplane as part of the SARP (Student Airborne Research Program) and the NASA G-III and G-V planes operating in the frame of STAQS. An overview of the measurements conducted by the three JPL lidars during this period is presented in Fig. 5.

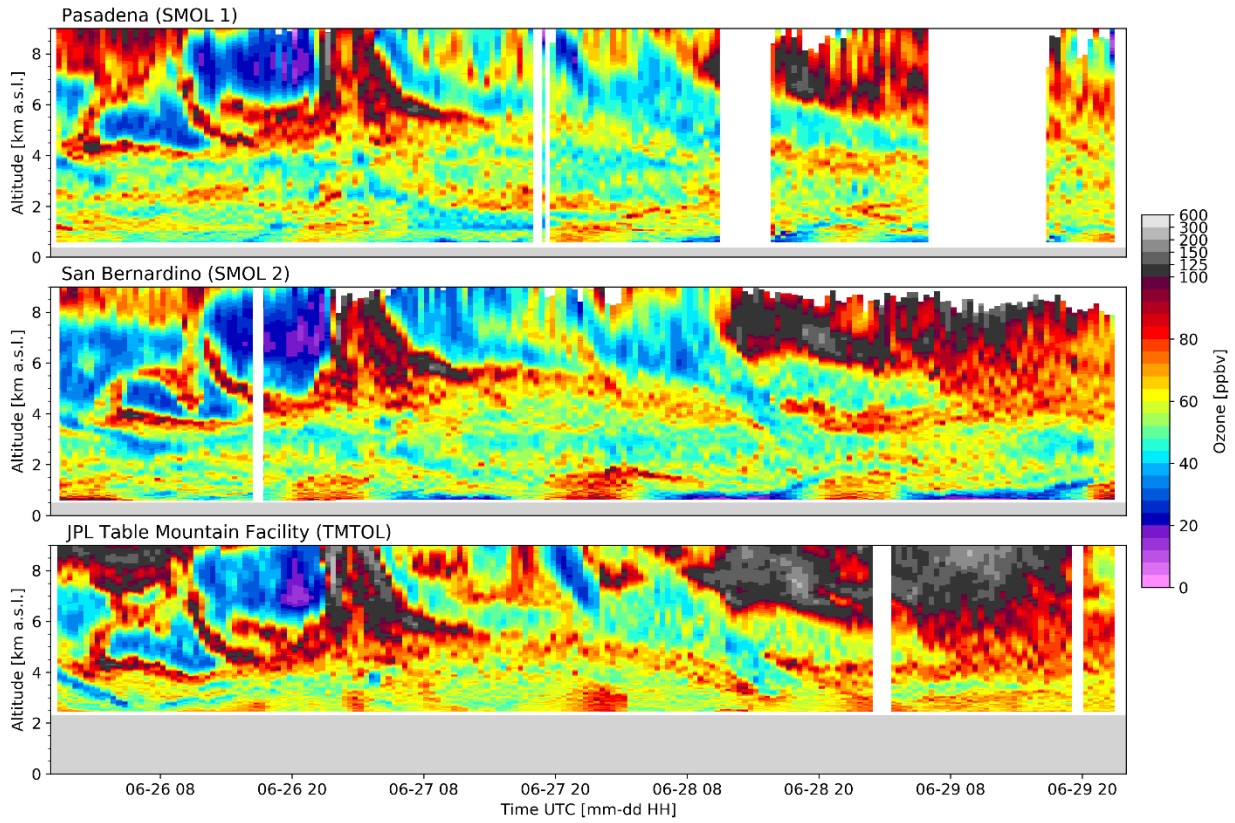

**Figure 5. SMOL-1 (top), SMOL-2 (middle), and TMTOL (bottom) measurements during IOP 1.**

The meteorological conditions during the first IOP were characterized by mostly clear skies at all observation sites, with temperatures slightly increasing over the five-day period. Some low-level clouds associated with the marine layer development affected the Pasadena site during the morning of June 28 and June 29 which created the data gaps shown in Fig. 5. As expected, due to the relatively short distance between the three observation sites, free tropospheric ozone (approx. above 3 km a.s.l.) exhibited very similar features, pattern and magnitude included, while the lowermost troposphere, reduced essentially to the Planetary Boundary Layer (PBL), shows substantial differences across the three observation sites.

The second IOP took place between 22 August and 27 August 2023, with TEMPO already in orbit. During this second part of the campaign, the NASA DC-8 participated in the frame of AEROMMA, and the G-III as part of STAQS. The payload of the airplanes remained the same when compared to the first IOP. Unfortunately, the NASA G-V airplane carrying the HSRL-2 Ozone wasn't available for the second IOP, which limited the availability of SMOL validation data. Figure 6 shows a curtain plot of the 5-continuous measurement days by the three JPL lidars during that second IOP. The skies were mostly clear, with almost no clouds throughout the entire period, which allowed to obtain nearly uninterrupted timeseries.

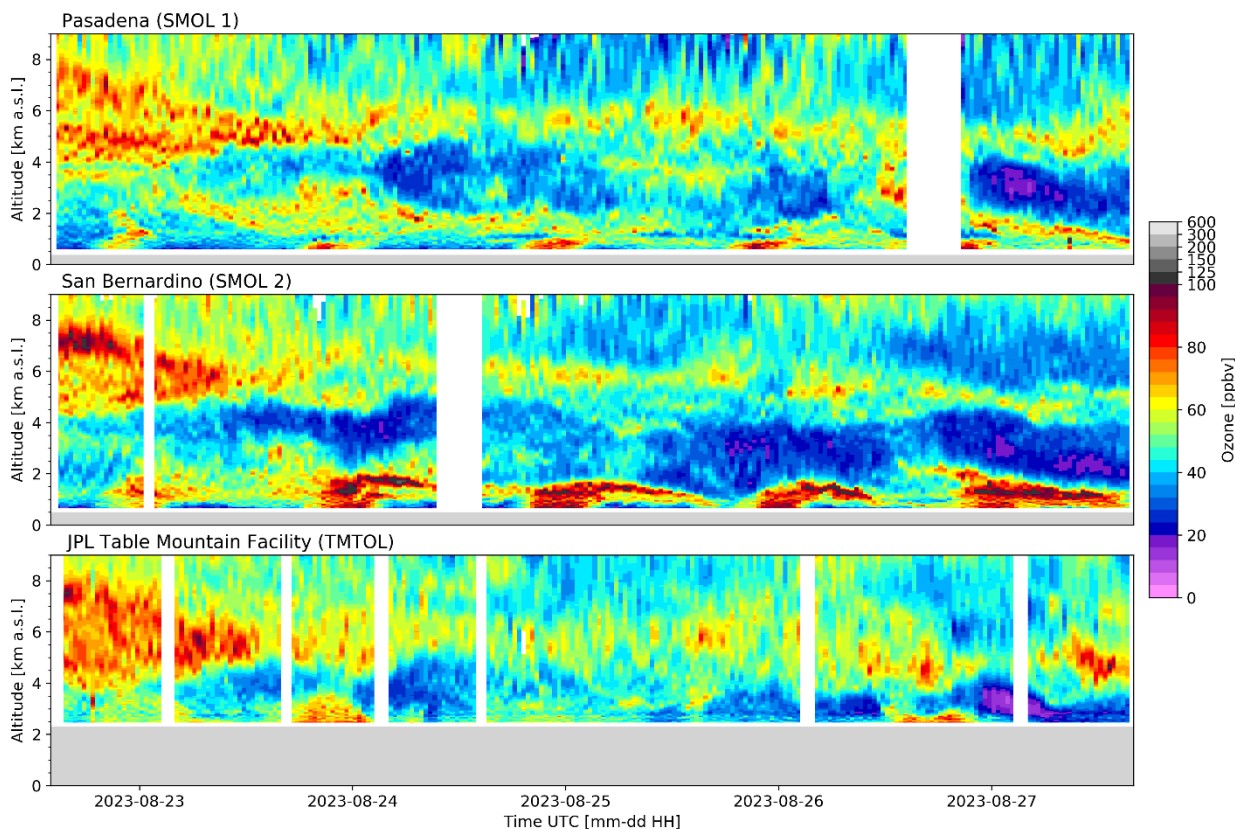

**Figure 6. SMOL-1 (top), SMOL-2 (middle), and TMTOL (bottom) measurements during IOP 2.**

### 4.1 Instrument validation

One of the main objectives of this first deployment was to validate the measurement capabilities of the SMOL lidars in the field, as well as to investigate the SMOL instruments stability and durability under relatively high ambient temperatures. The participation of many research organizations to the AEROMMA campaign and STAQS mission allowed a comparison of the SMOL measurements with several in-situ and remote sensing techniques from ground-based and airborne platforms.

#### 4.1.1 Comparison with airborne measurements

The NASA DC-8 hosted a large set of in-situ measurements for air quality, including ozone measurements from the NNOX and chemiluminescence (CL) instruments. The NOAA NNOx cavity ring-down spectrometer measured $NO_2$ directly, and $O_3$ via chemical conversion to $NO_2$, by absorption of light from a diode laser centered at 405 nm (Wild et al. 2014, Washenfelder et al. 2011). Air sampled through a ¼" OD Teflon inlet at ambient pressure split between the $NO_2$ and $O_x$ (= $O_3$ + $NO_2$) channels, with a flow rate of 1-2 VLPM through each. A brief overflow of the inlet with clean, dry air, every three minutes

during flight, provided instrument zeros devoid of absorbing species. Continuous addition of excess nitric oxide (NO) reagent gas (approximately $5\times10^{14}$ molecules cm$^{-3}$) to the $O_x$ channel quantitatively (>99%) converted ambient $O_3$ to $NO_2$ via NO +

$O_3 \rightarrow NO_2 + O_2$. Square-wave modulation of the diode laser at a frequency of 2 kHz induced exponential decay in light intensity within the optical cavity of each channel, which photomultiplier tubes detected. The difference in the time constant, or fit of this exponential decay, when absorbing species were present (ambient sampling) or absent (instrument zero), provided an absolute measurement of $NO_2$ number density in the channel. The primary source of uncertainty in the measurement ($\pm 12\%$) is the pressure- and flow-dependent effective absorption cross section of $NO_2$, which is a function of the length within the optical cavity over which the sample is present. The in-flight limit of detection during AEROMMA was 900 pptv. The NOAA Airborne Cavity Enhanced Spectrometer (ACES), which shared an inlet with NNOx during AEROMMA, also measured $NO_2$ spectroscopically and achieved better accuracy ($\pm 4\%$) and precision (50 pptv) than the NNOx $NO_2$ measurement (Min et al. 2016). For this reason, ACES $NO_2$ was subtracted from the NNOx $O_x$ measurement to yield the reported NNOx $O_3$. In situ ozone was also measured by chemiluminescence with pure nitric oxide (Cooper et al., 2024). Ambient ozone was measured at 10-Hz and reported as 1-second average. The in-flight precision was $\pm 50$ pptv and the total uncertainty was estimated to be $\pm 5\%$ (1-sigma).

The NASA G-V, available during the first IOP, carried the NASA Langley HSRL-2 Ozone lidar. This lidar provides aerosol backscatter and extinction profiles using the High Spectral Resolution Lidar technique at 355 and 532nm and backscatter profiles at 1064nm using the standard elastic backscatter technique. Particulate depolarization is measured at all three wavelengths (Hair et al., 2008). In addition, two ultraviolet wavelengths at 290.6 and 300.2nm are used for the Ozone DIAL measurements (Browell et al., 1998). The system has been compared to six ozonesondes launched during demonstration flights that resulted in a mean profile bias of -1.2% and a mean standard deviation of 5.7% for the profiles compared (Hair et al., 2018). In addition, comparisons have been made to both ground-based lidars and ozonesodes during the recent NASA Tracking Aerosol Convection Experiment – Air Quality (TRACER-AQ) that showed similar performance.

Both airplanes conducted several science flights over the LA area. The SMOL measurements were compared to the aircraft measurements made within a 25 km radius from the lidar sites, which provides a good compromise between the number of measurement coincidences and measurement representativeness. The results of these comparisons are summarized in Fig. 7. The mean of the datasets for all coincidences (Fig. 7, top row) gives a general idea of the ozone structure a the time of the overpasses. As expected, due to the variability in the terrain elevation, some of the airborne datasets extend past the ground level of the SMOL and TMTOL sites. This is especially true for the case of the HSRL datasets and their overpass over TMTOL.

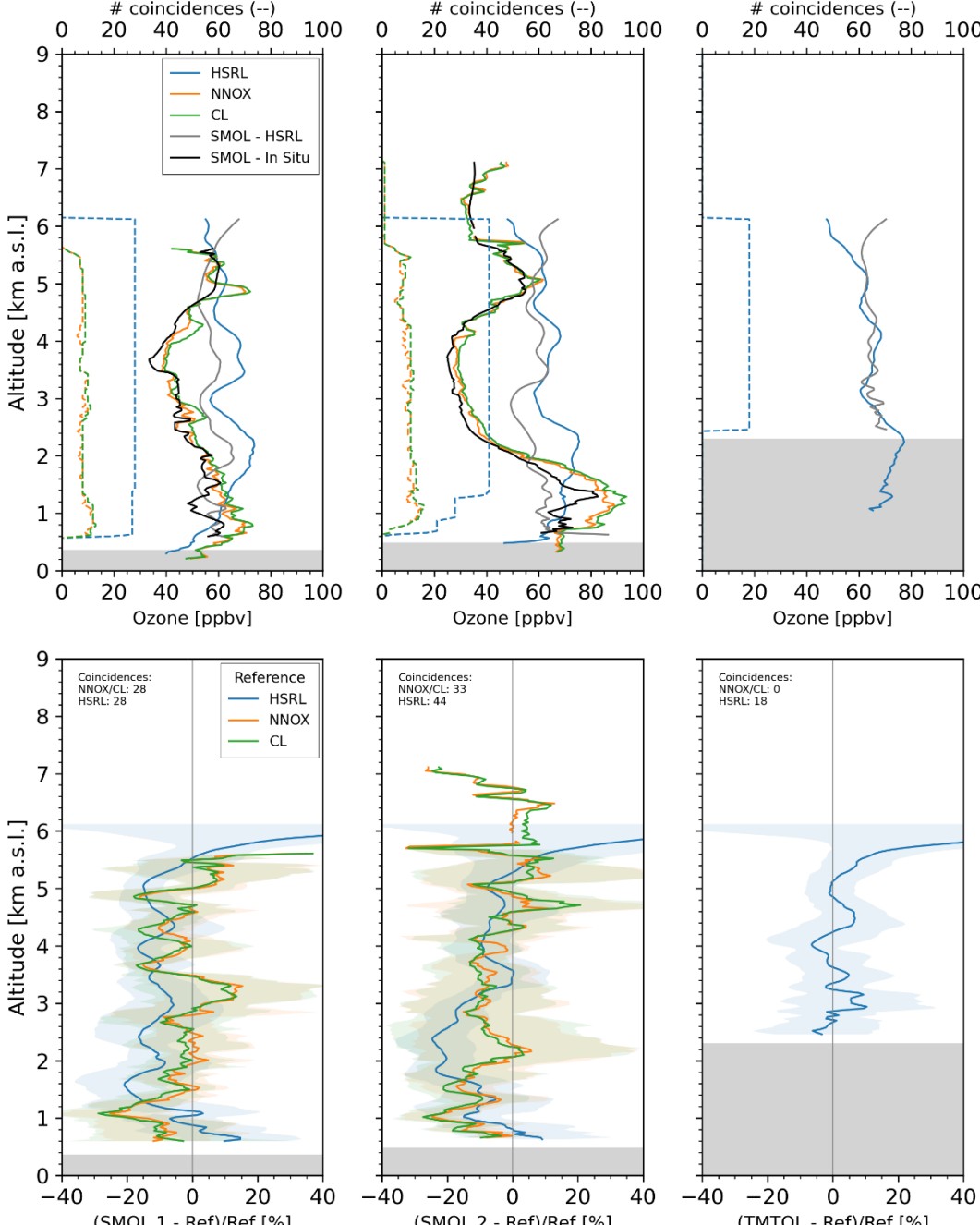

**Figure 7. SMOL/TMTOL comparison with airborne in-situ and lidar measurements based on the coincidence criteria shown in Fig.**
**4. The top row shows the mean of each dataset for all overpasses over SMOL-1 (left), SMOL-2 (middle), and TMTOL (right) ground-based sites. The number of coincident points between each airborne dataset and the ground sites is also shown as function of altitude (dashed). The bottom row shows the mean of the difference between each airborne dataset and SMOL-1 (left), SMOL-2 (middle),**

**and TMTOL (right). The standard deviation (1-sigma) for the difference between each airborne dataset and the ground based lidars is shown as shaded areas of the same color. The ground elevation for each site is show as grey shaded areas on each panel.**

In general, as shown in the bottom panels of Fig. 7, SMOL-1, SMOL-2, and TMTOL exhibit a good agreement with the airborne in-situ and lidar measurements across the whole 0.5-7 km altitude range. The SMOL instruments show the best agreement with the NNOX and CL in-situ measurements, with differences remaining within ±10 % at most altitudes, maximizing to 15-20% below 1.5 km (low bias). This latter discrepancy is likely due to a combination of aerosol contamination and ozone spatial variability in the PBL. The comparison of the SMOL instruments with the HSRL/DIAL shows a slight low

bias of about 10% across most of the measurement range, especially in the case of SMOL-1. A larger 15-20% low bias between SMOL-2 and the HSRL/DIAL over the 1.5 to 3 km altitude range is not present when comparing with the in-situ measurements. Finally, the HSRL shows a very good agreement with the TMTOL system up to 5 km a.s.l. Above that point, the airborne DIAL shows a large low bias when compared with all ground-based lidars. Further analysis reducing the coincidence criteria from 25 km to 6 km didn't show qualitative changes in the observed biases between the SMOL and the HSRL-2 Ozone lidar

(not shown), which suggests that the bias is likely not due to spatial ozone variability. The standard deviation of the difference, which is made up of a combination of measurement uncertainty of the compared techniques and spatial variability, exhibits a relatively constant magnitude for the altitude ranges where the number of coincidences is comparable. Since the measurement uncertainty of SMOL and the in-situ techniques are mostly constant with altitude, this constant behaviour suggests that spatial/temporal inhomogeneity is limited for these comparisons. At the bottom and top of the comparison range, the standard

deviation deviates due to the smaller number of coincidences. An extreme case for this is the comparison of the in-situ measurements with SMOL-2 (Fig. 7, middle column), where there is only one coincidence above 6.2 km, making the standard deviation zero.

### 4.1.2 Comparison with surface measurements

An important aspect of the SMOL system performance to be tested is its minimum measurement range, which is limited by a

combination of transmitter-detector field of view overlap and detector dynamic range. The SMOL instruments' lowest valid measurement points, which are typically between 150 and 250 m a.g.l., were compared to nearby surface measurements (Fig. 8). The surface data were obtained from nearby monitors operated by the the South Coast Air Quality Monitoring District (SC AQMD). The Pasadena monitor is about 8.5 km from the SMOL-1 deployment location, while the San Bernardino one is at 10.3 km from SMOL-2. The surface data is reported with a time resolution of 1 hour.

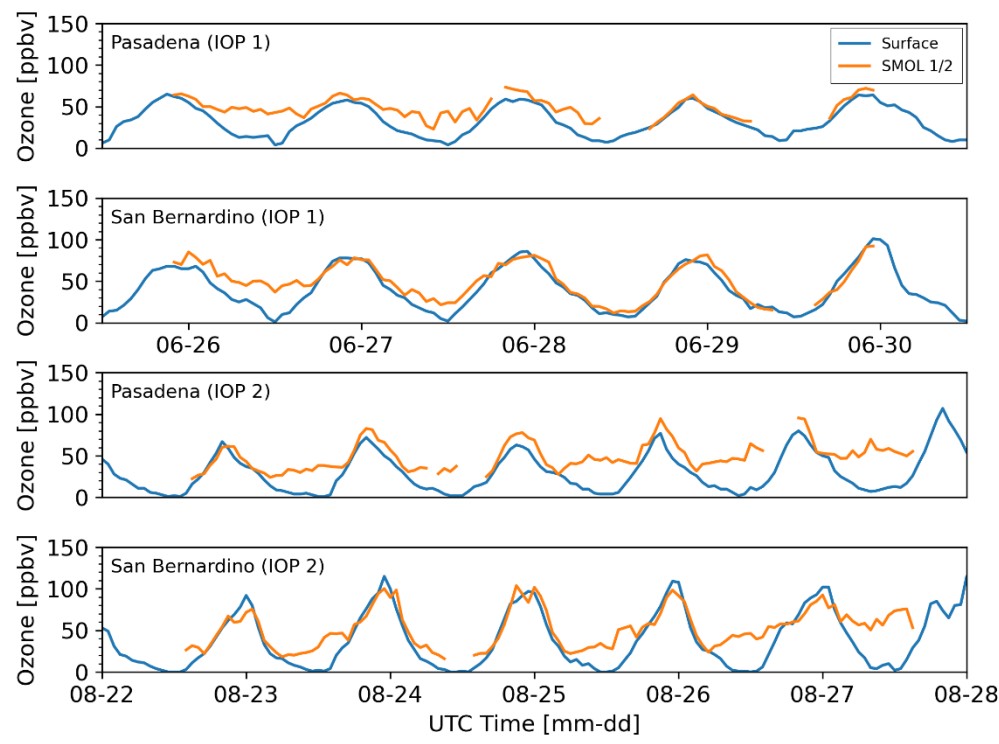


**Figure 8. Comparison of first valid SMOL data points with nearby surface ozone monitors during both IOPs.**

For both IOP 1 and IOP 2, the comparisons between SMOL and surface measurements show good agreement between the SMOLs' values measured at their lowest altitude and the surface values during daytime hours. This good agreement shows that the planetary boundary layer (PBL) is deep and well mixed during that time. After sunset however, as the boundary layer

top decreases and titration increases, a shallow layer of depleted ozone appears near the surface, giving place to a strong vertical ozone gradient over the first few hundred meters. This gradient was especially well pronounced during IOP 2 and causes a large difference between the surface measurements and the values measured by the SMOL instruments at their lowest available altitude.

### 4.1.3 Comparison with GEOS-CF

In this section we present a brief comparison of SMOL-2 measurements during the IOP 2 with the Goddard Earth Observing System Composition Forecast (GEOS-CF) model 'replay' results (Keller et al., 2021). While a full in-depth comparison between the two is beyond the scope of this work, these results provide further support for the need of continuous high resolution ozone observations that can be used for model validation and assimilation. The results presented in Fig. 9, corresponding to the IOP 2 in the San Bernardino area, indicate that GEOS-CF can reproduce a good fraction of the features

observed by SMOL 2, including ozone structures in the free troposphere, ozone build-up during the afternoon, as well as near-surface ozone depletion overnight. The timing of these features is also accurately captured by the model. On the other hand, a

quantitative comparison shows some over and underestimation by GEOS in the PBL ozone concentration, as well as some limitations to capture the fine structure of the ozone PBL distribution and residual layer.

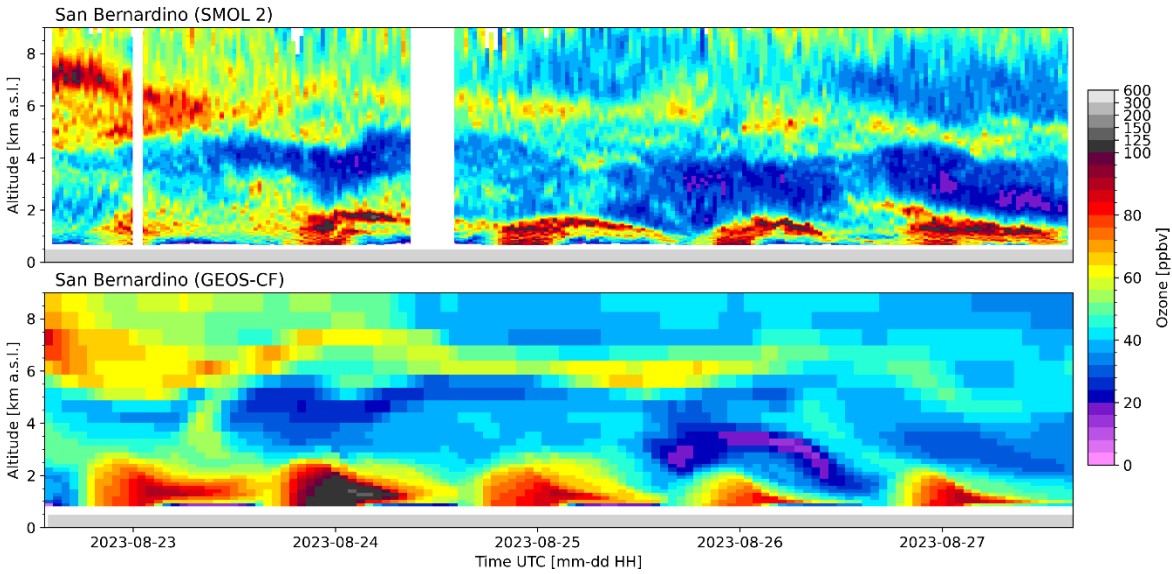

**Figure 9. Comparison of the results from SMOL-2 during IOP 2 and GEOS-CF. The upper panel shows the measurements by SMOL 2 during IOP 2, while the bottom panel shows the GEOS-CF replay results.**

## 5 Conclusions

As part of a recent effort to enhance our capability to measure tropospheric ozone, two new SMOL instruments were recently built at JPL and deployed on the field during summer 2023 and contributed measurements to the AEROMMA 2023 campaign

and NASA STAQS Mission.

While on the field, the two SMOL systems performed very well, with minimal downtime due to hardware failure. The measurement performance was within the initial specifications, with only a slight deterioration of STNR due to dust accumulation on top of the transmitter and receiver windows, and due to optics mild deterioration on the transmitter path. The system alignment remained stable throughout the observations. Some contamination on the interior of the Raman cell windows

was also observed, which was found to be caused by contamination during the cells manufacturing process. Potential future modifications to the system might reduce the transmitted power and use a dichroic beam splitter instead of the current intensity beam splitter to compensate for it.

The comparison with airborne in-situ and lidar measurements showed very good agreement considering the relatively coarse co-location criteria used for the comparisons and the high ozone spatial variability in the area.

The summer 2023 deployments have demonstrated that the development of affordable (below $100k USD), compact, and autonomous ozone lidars is feasible without compromising on the performance. The SMOL-1 and SMOL-2 performance is

similar to the other TOLNet instruments (Leblanc et al., 2018), with an accuracy better than 10% up to 8 km a.s.l. for 30-minute-resolution profile. Based on these results, we conclude that the SMOL instruments can become important actors in contributing to address current important science questions such as how physical, chemical, and dynamical processes impact air quality at local, regional, and continental scales, and at temporal scales of hours to decades.

**Data availability**

The SMOL and TMTOL data can be retrieved from https://tolnet.larc.nasa.gov/download (last access: 4 June 2024). All supporting measurements shown in this work can be provided by the corresponding authors upon request.

**Author contributions**

FC prepared most of the manuscript and conducted the statistical comparison with the rest of the datasets shown in this work. TL is the principal investigator of the JPL lidars and processed TMTOL and SMOL data shown in this manuscript. PW provided technical support for the collection of the data included in this work. SB, KZ, WC, CW, and JP provided the airborne in-situ data used in this study. JH and TS provided the airborne lidar data used in this study. JS contributed to the organization of STAQS and is the principal scientist of TOLNet. All co-authors provided feedback on the manuscript.

**Competing interests**

One of the co-authors is a member of the editorial board of AMT.

**Financial support**

The research was carried out at the Jet Propulsion Laboratory, California Institute of Technology, under a contract with the National Aeronautics and Space Administration (80NM0018D004). The authors acknowledge funding from the Tropospheric Chemistry Program of the NASA Earth Science Division.

*Acknowledgments* The authors would like to thank the South Coast Air Quality Management District for providing the surface ozone data used in this study.

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
