# Peer review of "The Small Mobile Ozone Lidar (SMOL): instrument description and first results"

_Atmospheric Measurement Techniques, 2024_

## Author Response (AR1)

Dear reviewers,

First, we would like to thank you for the comments and contributions to our manuscript. Below you can find the replies to your comments and changes made to the manuscript to address them.

**Anonymous Referee #1**

**General comments**

**i) The technical information on the system and data processing could be more detailed. The paper is submitted to AMT so technical details are in the scope of the journal and of interest to likely readers. Summary tables for technical specs of the transmitter, receiver, and counting electronics in an Appendix could be helpful for the reader.**

We included further technical details to the paper, including a table summarizing technical specs. Further details about the additions are presented in the specific comments section.

**ii) It could also benefit from a bit more discussion as to why you have chosen various hardware/optical components and some system efficiency calculations.**

Further discussion was added to the text. More details are provided in the specific comments section.

**iii) Introduction of scientific motivations could be stronger and more explicit. Why make these lidar measurements of tropospheric ozone above 250m? Is the objective that these instruments provide real time data to a weather forecast or reanalysis (i.e. ECMWF) or are these instruments only useful for CalVal of satellites/aircraft? Explicitly citing the work of a few other small network deployable lidars for water vapour, wind and aerosol could also be good.**

Further citations were included in the manuscript referencing other network deployable lidars: Engelman et al., 2016; Shimizu et al., 2016; Spuler et al., 2015.

**Specific comments**

**What are the other laser parameters? A measured histogram of laser pulse frequency over 30 minutes would give you an idea of what the stability of your laser is during a single ozone profile.**

Unfortunately, we don't have a spectrometer available that can monitor the laser stability at 266 nm with enough accuracy. Luckily, the wavelength stability is not as critical in this system as is in other DIALs that rely on very narrow absorption lines. SMOL operates in the Hartley absorption band, which doesn't have any complex structure in the absorption spectrum (see plot below from https://amt.copernicus.org/articles/12/569/2019/amt-12-569-2019.pdf).

[Figure]

For reference, even a large change of 0.1 nm on the transmitted wavelength will only have about 1% impact on the differential cross-section.

**Does your emission frequency drift over the day due to day/night heating of the instrument? Or due to the air conditioner turning on and off?**

We don't have a way to monitor the output frequency with enough accuracy, but we haven't observed any time dependent bias in the ozone products that can suggest such drift.

**How broad is your emission line before and after the gas cells? How does this compare to the target lines for O3 and the offline? Coupled with the histogram above, you can estimate how efficiently your laser pulse is illuminating the target O3 line(s) compared to the offlines.**

See previous answers.

**How is the collimation ensured? How well is this known? Is the beam shape OK after the gas cells? Is this why a wide fov is used in the receiver section?**

Collimation of the transmitter is roughly achieved first by adjusting the distance

between the focusing and collimating lens of the Raman cells based on the known characteristics of the lenses and wavelengths of operation. Once the system is approximately collimated, we scan the laser beam over the receiver FOV with the piezo actuators and make sure we have a plateau on this scan above the expected altitude of full overlap and no saturation (>2.5 km for the high-altitude receiver). The width of the plateau is related to the combination of the receiver FOV and transmitted divergence. We adjust the receiver FOV (fiber position) and transmitted divergence (recollimation lens position) until we approximately achieve the expected plateau width. Having wide FOVs on the receivers is one of the luxuries we can allow ourselves by operating is mostly solar blind region of the spectrum and makes the whole process less stringent. The following was added to the manuscript:

*"The recollimation of the transmitted beam is verified by scanning the beam over the field of view (FOV) of the receivers and looking for an intensity plateau on these scans at given altitude when full overlap and no saturation is expected (>2.5 km for the high range receiver). The angular width of this plateau is a combination of the beam divergence and the receiver FOV."*

**typo L94: "transmitted through..."**

Corrected.

**L97: Nominal Ocular Hazard Distance not defined**

Added.

**Is there a benefit to choosing a smaller NA on the fibres to reduce the field of view further and reduce the influence of the solar background even more, or is the lidar already sufficiently solar blind? Some diagnostic plots could be helpful. Later in the paper, comments are made about complications with background estimations.**

While reducing the FOV could slightly reduce the background, specially at 299 nm, the gains will be small, as the instrument range is mostly limited by the absorption of the 'on' wavelengths (289 nm and 266 nm). Furthermore, such reduction in the FOV of the receivers would make the instrument more sensitive to misalignment during transportation and operation.
We also noted a mistake on the paper for the diameter of the fibers used in the medium and low range channels. The diameter of the fibers was 0.4 mm and not 1 mm and 0.2 mm.

The following was added to the manuscript to explain this design decision:

*"The resulting field of view (FOV) of receivers are 1.3 mrad for the high range and 4 mrad for the medium and low range receivers. While smaller FOVs could help to reduce the impact of solar background on the 299 nm channels during daytime operation (266 nm and 289 nm are practically solar blind), the instrument range is mostly limited by the on-wavelength absorption. Furthermore, such change would make the instrument more sensitive to misalignment caused by temperature changes, vibration, laser pointing jitter, etc."*

**What is your estimated beam spot size inside your field of view?  How much can the beam jitter and drift inside the fov?   Do you have stabilization for day/night observations to account for thermal expansion of the lidar unit and slow diurnal misalignments?  What about daytime turbulence corrections?  Local convection can jitter your beam a lot inside the fov on the timescales of 1 second to minutes, reducing signal stability.   Where you place the SMOL could also be important for the signal stability (grassy field vs. concrete building roof, ground level, vs. elevated platform etc)**

In the case of the high range receiver (the one with the narrowest field of view) the FOV is 1.3mrad, which is about 1.3 m at 1 km. The beams are 0.25 mrad, which is 25 cm at 1 km. The large field of view of the receivers compared with the transmitted beams makes the system overall very stable and, based on our experience, unsensitive to internal temperature gradients or turbulence.

**L110:  Is FWHM of 32 nm a typo?**

No, it is not a typo. The first filter (FWHM 32 nm, SP in the diagram) is used to provide additional background suppression on the receiver. Such wide filter allows to encompass the two wavelengths of the receiver (289 nm and 299 nm), while providing additional solar background blocking on wavelengths above 310 nm.

**If you are throwing away half your signal with the intensity beam splitters, why not just use narrower filters?  Even if the transmission efficiency of a cheaper deep UV filter is low (~30%), isn't it better to have a 30% intensity signal at a well known, narrow line than a 50% signal over a broad wavelength range?**

**Saturation problems.  It would be good to have more details.  This is a bit vague and the reader can't tell how impactful the problem is in the current design.  A**

**signal plot could be helpful.  Is there a benefit to increasing the rep rate of your laser from 20Hz and increasing the speed of acquisitions?**

The saturation level of the receiver is given by the PMTs selected for this design. These PMTs have built-in discriminators with a pulse pair resolution of 20 ns, which give a maximum nominal count rate of 50 MHz. Well before reaching that point, the PMTs are affected by pile-up. Following the literature and the manufacturer recommendation, a correction is applied (see plot below from PMT datasheet https://www.hamamatsu.com/content/dam/hamamatsu-photonics/sites/documents/99_SALES_LIBRARY/etd/H12386_TPMO1073E.pdf).

[Figure]

Figure 5: Count rate linearity correction

This correction allows to extend the usable range from ~1MHz to ~10MHz.

While the selection of this PMT type has some disadvantages (20 ns pulse pair resolution vs 4-5 ns of other more common PMTs), having a built-in discriminator reduces the chance of electrical noise affecting the PMT signal between the PMT and the discriminator and simplifies the overall design by not requiring a custom-made discriminator.

The following was added to the manuscript:

*"While these photomultipliers are relatively slow (20 ns pulse pair resolution) compared to other PMTs used by other lidars at JPL TMF, the fact that they have a built-in discriminator in the same package as the PMT minimizes the chance of electrical noise from the laser and other subsystems to impact the signal. Furthermore, this also simplifies the design and eliminated the need to further adjust the discriminator level as this is made in the factory to optimize the detector performance."*

**What are the technical specification of the MCS and counters?  Dark count rate? Quantum efficiency? Counter clock rate?  Data write rate?**

Further information about the counter and PMTs was added to the system specification table. Also, the data write rate was included to the receiver section text: *"(every 3 minutes)".*

**If the alignment is at a set time, how do you account for variable sky conditions?  For example, a patchy cirrus moving overhead could lead to a variable maximum back scatter due to cloud structure and result in a poor alignment.**

The alignment schedule is manually input based on the expected meteorological conditions. Since the system can operate several days without realignment, we can wait to have good weather conditions to perform this realignments/alignment checks.

**Running the piezios every second could give you real-time beam stabilization to compensate for daytime turbulence.  Disabled during cloud detection.**

We are considering the pros and cons (basically more complexity) of having a more automated realignment. So far, we haven't seen the need for it, but this might change as more units are deployed.

**Non-linear backgrounds are an unfortunate reality for some historical lidar datasets.  But this should be corrected in hardware for newly designed systems.  Showing some plots of lidar returns at different signal intensities, could show the linear range for SMOL1 and 2.**

An example of the SMOL-1 lidar signals (basically the same for SMOL-2) is now included in the manuscript.

[Figure]

*Figure 3. Raw signals of SMOL-1 during AEROMMA averaged over 30 minutes.*

We are currently working on further reducing the impact of the SIN in the 266 nm channel and we hope to minimize the need for correction as we upgrade the systems.

The following text was added for clarification:

*The most obvious case of SIN can be seen on the 266 nm channel on Fig. 3. While 266 nm is almost completely solar blind due to the strong ozone absorption at this wavelength, the background on this channel is far from zero and is not constant with range, which is an indication of SIN. In this case, SIN can be modelled as a series of two exponentials.*

**Is vertical resolution in GLASS achieved through integration or filtering.  If filtering, what filter?**

GLASS retrieval uses a derivative Savitsky Golay filter followed by a Blackman filter. GLASS can operate in different modes, including fixed vertical resolution and constant random uncertainty. In the case of the data presented in this work, the second option is used. The software automatically adjusts the vertical resolution of the filter to achieve a constant random uncertainty.

The following clarification was added to the text:

*"...with the derivative step implemented through a Savitsky-Golay (SG) derivative filter followed by a Blackman filter for additional noise reduction."*

*"...(controlled by the length of the SG and Blackman filter windows)."*

**It seemed to me that there were more technical specifications and error tolerances given for the HSRL than the SMOL system.  This comes back to my main point that more details of the new instrument would be appreciated.**

We added further information about the system, hopefully that is enough to provide a more comprehensive explanation of some of the design decisions.

**Figure 6. could include the 2 sigma limit on the mean. Seeing the measured ozone profiles might also be nice, perhaps a 2-panel plot.**

Further information was added to the figure. The top row shows now the average of the overpasses for each instrument as well as the corresponding average of the SMOL measurements. Additionally, we included the number of coincident points as function

of altitude, mostly relevant for the in-situ measurements, as those typically those do not cover the whole SMOL altitude range.

Finally, we also included the 1 sigma standard deviation of the difference values used to calculate the mean. Since the number of values used to calculate the mean change with altitude, the interpretation of this value might not be completely clear. For example, in the case of the middle panel above 6.2 km the standard deviation is not defined because there is only one coincidence between the in-situ and SMOL2 measurements.

The original figure was replaced by:

[Figure]

*Figure 7. SMOL/TMTOL comparison with airborne in-situ and lidar measurements based on the coincidence criteria shown in Fig. 4. The top row shows the mean of each dataset for all overpasses over SMOL-1 (left), SMOL-2 (middle), and TMTOL (right)*

*ground-based sites. The number of coincident points between each airborne dataset and the ground sites is also shown as function of altitude (dashed). The bottom row shows the mean of the difference between each airborne dataset and SMOL-1 (left), SMOL-2 (middle), and TMTOL (right). The standard deviation (1-sigma) for the difference between each airborne dataset and the ground based lidars is shown as shaded areas of the same color. The ground elevation for each site is show as grey shaded areas on each panel.*

The following text sections were also added to the manuscript to further discuss the additional information provided by the updated figure:

*"The results of these comparisons are summarized in Fig. 7. The mean of the datasets for all coincidences (Fig. 7, top row) gives a general idea of the ozone structure a the time of the overpasses. As expected, due to the variability in the terrain elevation, some of the airborne datasets extend past the ground level of the SMOL and TMTOL sites. This is especially true for the case of the HSRL datasets and their overpass over TMTOL."*

*"The standard deviation of the difference, which is made up of a combination of measurement uncertainty of the compared techniques and spatial variability, exhibits a relatively constant magnitude for the altitude ranges where the number of coincidences is comparable. Since the measurement uncertainty of SMOL and the in-situ techniques are mostly constant with altitude, this constant behavior suggests that spatial/temporal inhomogeneity is limited for these comparisons. At the bottom and top of the comparison range, the standard deviation deviates due to the smaller number of coincidences. An extreme case for this is the comparison of the in-situ measurements with SMOL-2 (Fig. 7, middle column), where there is only one coincidence above 6.2 km, making the standard deviation zero."*

**Comparison with a reanalysis or forecast (ECMWF) seems relatively straight forward to include, and could provide more motivation for the importance of high resolution observations to capture small-scale features at a local level.**

The following section was added to the manuscript.

**4.1.3 Comparison with GEOS-CF**

*In this section we present a brief comparison of SMOL-2 measurements during the IOP 2 with the Goddard Earth Observing System Composition Forecast (GEOS-CF) model 'replay' results (Keller et al., 2021). While a full in-depth comparison between the two is beyond the scope of this work, these results provide further support for the need of continuous high resolution ozone observations that can be used for model validation and assimilation. The*

*results presented in Fig. 9, corresponding to the IOP 2 in the San Bernardino area, indicate that GEOS-CF can reproduce a good fraction of the features observed by SMOL 2, including ozone structures in the free troposphere, ozone build-up during the afternoon, as well as near-surface ozone depletion overnight. The timing of these features is also accurately captured by the model. On the other hand, a quantitative comparison shows some over and underestimation by GEOS in the PBL ozone concentration, as well as some limitations to capture the fine structure of the ozone PBL distribution and residual layer.*

[Figure]

**Figure 9. Comparison of the results from SMOL-2 during IOP 2 and GEOS-CF. The upper panel shows the measurements by SMOL 2 during IOP 2, while the bottom panel shows the GEOS-CF replay results.**

**Anonymous referee # 2**

**Specific comments**

**Abstract line 16: change 'physical' to 'physical and chemical'.**

Done.

**Abstract line 23: it says three ozone DIAL pairs, but the listing is ambiguous. Please clarify. One DIAL pair is applied twice, but perhaps this is better left to the descriptions later**

The sentence was rephrased as: *"...three ozone DIAL pairs, including one 266/289 and two 289/299 nm."*

**Introduction line 60: what is the expected bias? Is that specified in TOLNET, before and after corrections for e.g. aerosol interference? Please add some text to explain this.**

**Instrument description line 90: How is the output energy for 266 nm optimised and stabilised and what is the (long term) stability without adjustment of the fourth harmonic crystal? Is the laser output power monitored and can it be optimised remotely?**

The following sentence was added to the section:

*"The laser output energy is stable to within ± 10% for multiple days without needing readjustment. As the flashlamps deteriorate and power decreases, remote adjustments to the flashlamp voltage and temperature adjustments to the doubling and quadrupling crystal efficiency allow to partially offset the power decrease and extend the service intervals."*

**Instrument description line 97: what is NOHD?**

A clarification was added to the text.

**Instrument description line 117: potential future updates might be moved to a summarising section, or the conclusions.**

This line *"Potential future modifications to the system might reduce the transmitted power and use a dichroic beam splitter instead of the current intensity beam splitter to compensate for it."* was moved to the conclusions section.

**Instrument description line 128: which parameters are recorded needed for the retrieval?**

A clarification was added to the text: *"(system location, elevation, and bin number corresponding to the zero range)."*

**Instrument description Figure 2: in the figure the wavelengths at output and detection are missing. Please add.**

Done.

**Instrument description Figure 2: a second figure will be helpful to indicate various control sub-system units as described in Sec.2.3**

Figure 2 was upgraded to include more information about the sub-systems described in Sec. 2.3.

[Figure]

**Figure 2. Schematic of SMOL. MM is multi-mode, SP is short pass, BS is beam splitter, IF is interference filter, MCS stands for multi-channel scaler, PDU is a power distribution unit, and UPS is an uninterruptible power supply.**

**Data processing Line 183: use '… specified optical thickness'.**

Done.

**References: perhaps a few older references can be selected to indicate earlier attempts for routine monitoring of tropospheric ozone profiles, in particular from the European Eurotrac TESLAS and TOR programmes. E.g.**
- **https://doi.org/10.5194/amt-13-6357-2020**

- **TESLAS: Tropospheric Environmental Studies by Laser Sounding (TESLAS), in: Transport and Chemical Transformation of Pollutants in the Troposphere, Vol. 8, Instrument Development for Atmospheric Research and Monitoring, edited by: Bösenberg, J.,Brassington, D., and Simon, P. C., Springer (Berlin, Heidelberg, New York), ISBN 3-540-62516-X, 1–203, 1997.**

A reference to these previous efforts was included in the manuscript:

*"The SMOL design leverages on the lessons learned from earlier attempts to established continued ozone monitoring in the troposphere (Bösenberg, 2000; Trickl et al., 2020), as well as over two decades..."*